# Disruption of Transmembrane Phosphatidylserine Asymmetry by HIV-1 Incorporated SERINC5 Is Not Responsible for Virus Restriction

**DOI:** 10.3390/biom14050570

**Published:** 2024-05-10

**Authors:** Gokul Raghunath, Elizabeth H. Abbott, Mariana Marin, Hui Wu, Judith Mary Reyes Ballista, Melinda A. Brindley, Gregory B. Melikyan

**Affiliations:** 1Department of Pediatrics, Division of Infectious Diseases, School of Medicine, Emory University, Atlanta, GA 30322, USA; graghun@emory.edu (G.R.); mmarin@emory.edu (M.M.); hwu8@emory.edu (H.W.); 2Children’s Healthcare of Atlanta, Atlanta, GA 30322, USA; 3Emory College of Arts and Sciences, Emory University, Atlanta, GA 30322, USA; 4Department of Infectious Diseases, College of Veterinary Medicine, University of Georgia, Athens, GA 30602, USA; jreyesb@uga.edu (J.M.R.B.); mbrindle@uga.edu (M.A.B.); 5Department of Population Health, College of Veterinary Medicine, University of Georgia, Athens, GA 30602, USA

**Keywords:** SERINC, virus restriction factor, flip-flop, Annexin V, lipid asymmetry, lipid exchange, scramblase, cyclodextrin

## Abstract

Host restriction factor SERINC5 (SER5) incorporates into the HIV-1 membrane and inhibits infectivity by a poorly understood mechanism. Recently, SER5 was found to exhibit scramblase-like activity leading to the externalization of phosphatidylserine (PS) on the viral surface, which has been proposed to be responsible for SER5’s antiviral activity. This and other reports that document modulation of HIV-1 infectivity by viral lipid composition prompted us to investigate the role of PS in regulating SER5-mediated HIV-1 restriction. First, we show that the level of SER5 incorporation into virions correlates with an increase in PS levels in the outer leaflet of the viral membrane. We developed an assay to estimate the PS distribution across the viral membrane and found that SER5, but not SER2, which lacks antiviral activity, abrogates PS asymmetry by externalizing this lipid. Second, SER5 incorporation diminished the infectivity of pseudoviruses produced from cells lacking a flippase subunit CDC50a and, therefore, exhibited a higher baseline level of surface-accessible PS. Finally, exogenous manipulation of the viral PS levels utilizing methyl-alpha-cyclodextrin revealed a lack of correlation between external PS and virion infectivity. Taken together, our study implies that the increased PS exposure to SER5-containing virions itself is not directly linked to HIV-1 restriction.

## 1. Introduction

The serine incorporator protein (SERINC) family encodes multi-pass transmembrane proteins that are ubiquitously expressed across various eukaryotic cells [1]. These proteins are believed to facilitate the transfer and incorporation of serine into phospholipids at the membrane interface and are thought to play an important role in sphingolipid biosynthesis [1]. The SERINC family of proteins are highly conserved and share significant sequence homology with each other [2]. The discovery of these proteins within the HIV-1 lipid envelope has inspired several studies that identified these proteins as potent host restriction factors inhibiting HIV-1 infection [3,4,5,6,7,8,9]. Despite sequence homology between the family members, SERINC5 (SER5) displays the highest restriction activity, whereas SERINC3 (SER3) and SERINC2 (SER2) exhibit moderate and no restriction activity, respectively [10]. Besides HIV-1, recent reports indicate that SER5 can also restrict other viruses, such as Murine Leukemia Virus (MLV) [6], Influenza A viruses [11], Hepatitis B virus [12], and SARS-CoV-2 [13]. The antiviral activity of SER5 is antagonized by different viral proteins, particularly by the HIV-1 Nef, the MLV GlycoGag protein, and the S2 protein from the Equine Infectious Anemia Virus [3,4,14,15,16]. The restriction potency of SER5 is envelope glycoprotein dependent, with lab-adapted strains of HIV-1 envelope glycoprotein (Env) being more sensitive to SER5 restriction than primary isolates [17,18,19]. Given the ability of SER5 to restrict multiple lentiviruses and the ability of different viral proteins to antagonize restriction, the fundamental mechanism of SER5 restriction has been a topic of significant interest to many.

Of particular interest is the question of how SER5 disrupts the fusogenicity of envelope glycoproteins. While several key details regarding the mechanism remain unknown, it is widely believed that the Env conformation is altered when SER5 is incorporated into the viral membrane, as evidenced by exposure to cryptic Env epitopes [16,17]. We have also shown that SER5 causes accelerated functional inactivation of Env over time and blocks HIV-1 fusion at a pre-hemifusion stage [18,20,21]. By comparison, cryo-electron microscopy (cryo-EM) studies suggest that membrane fusion is arrested at a small pore stage, which indicates that SER5 can also block the late steps of viral fusion [22]. Despite considerable efforts, it is still unclear if SER5 directly interacts with Env. Whereas Zhang et al. reported direct Env-SER5 binding using co-IP and bimolecular fluorescence complementation assays [23], our super-resolution imaging experiment did not reveal the colocalization of these proteins on HIV-1 pseudoviruses [24].

An indirect mechanism that involves local modulation of lipid membrane properties was proposed by us and others [24,25], but the exact nature of such disruption remains poorly understood. Using quantitative lipidomics, Trautz and colleagues showed that the overall lipid composition of neither the producer cells nor the HIV-1 particles was altered due to SER5 expression [26]. We have previously reported, using Nile Red and Laurdan probes, that disruption of the global lipid order of SERINC-containing virions does not correlate strongly with its restriction potential [27]. However, cryo-EM imaging combined with a membrane tension probe Flipper-TR suggested that SER5 modulates the lipid phase structure of the viral envelope and thus can affect viral fusion indirectly [25]. Interestingly, the authors also found that the addition of exogenous phosphatidylethanolamine (PE) can antagonize restriction by SER5, whereas the addition of neither phosphatidylcholine (PC) nor phosphatidylserine (PS) led to significant changes to viral particle fusogenicity. These results imply that the restriction mechanism of SER5 may depend on its ability to alter the organization of the viral lipid envelope, without changing its overall composition.

A more recent report by Leonhardt et al. showed that both SER5 and SER3 can act as lipid transporters between membrane leaflets when incorporated into virions, thus disrupting membrane asymmetry [28]. By utilizing a PS-sensitive probe, Annexin V (AnxV), alongside ultrastructural studies, they were able to show that SERINC incorporation into both HIV-1 and MLV leads to PS externalization, correlating strongly with loss of infectivity. The purported increase in PS levels on the viral surface indeed is a striking result for many reasons. First, spontaneous phospholipid flip-flop rates in membranes are known to be very slow [29,30,31,32]. Second, it has been shown that the diffusivity of lipids is generally very slow in vesicles that mimic viral membrane compositions both in terms of lateral mobility [33] and trans-bilayer flip-flop rates [34,35]. Finally, there are no known viral proteins that can actively maintain or modulate lipid asymmetry after budding from the producer cells. Hence, the ability of SER5 to modulate trans-bilayer lipid asymmetry on viruses might be a critical aspect of its restriction mechanism. However, it is unclear if the loss of PS asymmetry across the HIV-1 membrane (in other words, externalization of viral PS) is the root cause of reduced infectivity of SER5-containing virions or a consequence of a more complex underlying mechanism.

Several viruses, like Vaccinia, Dengue, Chikungunya, and Ebola, are known to hijack the cellular machinery to externalize the PS on infected cells, and, as a result, increase the PS level on the surface of budding virions; this, in turn, can promote viral attachment to PS receptors and internalization by cells (referred to as apoptotic mimicry) [36,37]. There are also reports that viral PS can act as a cofactor for HIV-1 infection [38]. In contrast, the exogenous addition of PE but not PS to HIV-1 in the presence or absence of SER5 has been reported to alter the fusogenicity of the viruses [25]. At a target cell level, it has been shown that HIV-1 entry and, as a result, infection is linked to the virus-induced externalization of PS [39]. Taken together, these studies hint that the interplay between PS asymmetry and SERINC incorporation is likely very complex and, therefore, warrants a thorough investigation of the underlying mechanisms.

To address this gap in knowledge, we sought to probe if disruption of PS asymmetry plays a significant role in the HIV-1 restriction mechanism of SER5. We utilized a multi-pronged experimental approach, including infectivity assays and single HIV-1 pseudovirus imaging to quantitatively analyze PS externalization on a per-particle basis. We found that the incorporation of SER5 into the virions leads to increased PS exposure at levels comparable to virions containing the non-specific scramblase mutant, TMEM16F, but that scramblase incorporation was not as deleterious for infectivity as SER5. Through endogenous and exogenous manipulation of viral lipids, utilizing producer cells lacking cell division cycle protein 50 (CDC50a) and methyl-α-cyclodextrin to exchange lipids with the viral membrane, we show that the changes in PS exposure on SER5-containing virions do not correlate with its restriction potency. Taken together, our results show that HIV-1 infectivity and SER5 restriction are largely unrelated to the disruption of PS asymmetry across the virus membrane. It is thus likely that PS externalization is a correlate but not a primary mechanism of SER5-mediated restriction, hinting at a more complex mechanism at play.

## 2. Materials and Methods

### 2.1. Cells, Chemicals, Lipids, and Plasmids

The following reagents were obtained through the NIH HIV Reagent Program, Division of AIDS, NIAID, NIH: TZM-bl cells, ARP-8129, contributed by Dr. John C. Kappes, Dr. Xiaoyun Wu and Tranzyme Inc; Saquinavir, ARP-4658, contributed by DAIDS/NIAID; and Polyclonal Anti-Human Immunodeficiency Virus Immune Globulin, Pooled Inactivated Human Sera, ARP-3957, contributed by NABI and National Heart Lung and Blood Institute (Dr. Luiz Barbosa). All cells were grown in a high-glucose DMEM with (Corning, Glendale, AZ, USA) or without phenol red (Cytiva, Marlborough, MA, USA) supplemented with 10% heat-inactivated fetal bovine serum (FBS, Atlanta Biologicals, Flowery Branch, GA, USA) unless otherwise specified. All media were supplemented with 100 units/mL penicillin/streptomycin antibiotics (Gemini Bio-Products, Sacramento, CA, USA). HEK293T/17 cells were obtained from ATCC (Manassas, VA, USA). All HEK293T cells were grown in the DMEM-based medium and supplemented with 0.5 mg/mL of Geneticin (G418, Cellgro, Mediatech, Manassas, VA, USA). For the generation of CDC50a KO cells, we utilized three guide RNAs (TACGGCTGGCACGGTGCTAC, TCGTCGTTACGTGAAATCTC, and GTGAACTGGCTTAAACCAGT), which were inserted into pSpCas9(BB)-2A-GFP (pX458), a gift from Feng Zhang (Addgene plasmid #48138, RRID: Addgene_48138) [40]. 293T cells were transfected in 6-well plates with equivalent amounts of plasmids encoding each gRNA using Polyplus Jet Prime (Polyplus-transfection, Illkirch-Graffenstaden, France) following the manufacturer’s instructions. Two days after transfection, the cells were counted and plated at a density of 0.5 cells/well in 96-well plates. The cells were monitored for 3 weeks to maintain single colony clones and non-clonal wells were discarded. Wells corresponding to single clones were expanded to 24-well plates and assessed for CRISPR knockout. Genomic DNA was extracted, and PCR was used to amplify the gRNA regions.

A Lenti-X concentrator was obtained from Takara Bio (San Jose, CA, USA). A 1 M solution of 4-(2-hydroxyethyl)-1-piperazineethanesulfonic acid (HEPES) was obtained from Cytiva. A 10× Annexin V binding buffer was purchased from BD Biosciences (Franklin Lakes, NJ, USA). Infectivity measurements were performed with the aid of a Bright-Glo luciferase kit (Promega, Madison, WI, USA). Poly-L-lysine, poly-D-lysine, Methyl-β-cyclodextrin, and Streptolysin O (from streptococcus pyogenes) were all purchased from Sigma-Aldrich (St. Louis, MO, USA). Methyl-α-cyclodextrin was purchased from AraChem (Tilburg, The Netherlands). Dithiothreitol (DTT) was purchased from American Bioanalytical (Canton, MA, USA). Membrane order probe Laurdan (6-Dodecanoyl-2-Dimethylaminonaphthalene), 16% Paraformaldehyde stock, Annexin V–Alexa Fluor 647, and goat anti-mouse Alexa Fluor 568 were obtained from ThermoFisher Scientific (Waltham, MA, USA). While we did not extensively validate this result, we observed that Alexa Fluor 647 labeled Annexin V purchased from a different manufacturer did not stain viral PS as effectively. Anti-HA.11 (cat #901501) antibody was purchased from BioLegend (San Diego, CA, USA). Rabbit anti-mouse IgG HRP was purchased from EMD Millipore (St. Louis, MO, USA, cat. No. AP160P). Goat anti-human IgG HRP was purchased from ThermoFisher Scientific (Waltham, MA, USA, cat. no. 31412) and mouse anti-GAPDH was sourced from Proteintech (Rosemont, IL, USA).

All lipids, including 1, 2-dioleoyl-sn-glycero-3-phosphocholine (DOPC), 1,2-dioleoyl-sn-glycero-3-phospho-L-serine (sodium salt) (DOPS), 1,2-dioleoyl-sn-glycero-3-phosphoethanolamine (DOPE), Cholesterol (Chol), 1,2-dioleoyl-sn-glycero-3-phosphoethanolamine-N-(7-nitro-2-1,3-benzoxadiazol-4-yl) (NBD-PE), 1,2-dioleoyl-sn-glycero-3-phosphoethanolamine-N-(cap biotinyl) (Biotin-PE), and Sphingomyelin (SM) from porcine brain, were purchased from Avanti Polar Lipids (Alabaster, AL, USA).

The pCAGGS plasmid encoding SER5-sensitive HIV-1 HXB2 Env, pcRev plasmids, CMV-SER5-iHA, CMV-SER2-iHA, packaging vector pR9ΔEnvΔNef, SER5-resistant JR2 envelope glycoprotein, and pCDNA3.1 vector have been described in prior publications [24,41]. Plasmids encoding core-fluorescent markers such as GFP-Vpr and mRFP-Vpr construct were described previously [42]. For efficient mTMEM16F incorporation in pseudoviruses, we utilized mTMEM16F D430G-FLAG, which was generously provided by Dr. Walther Mothes [28,43].

### 2.2. Virus Production, Purification, and Western Blotting

HIV-1pps were produced by transfecting HEK293T/17 or ΔCDC50a-HEK293T cells using Jet PRIME transfection reagent. Viruses were produced from transfecting cells seeded either in 10 cm dishes or in 6-well culture plates, respectively. In both cases, the transfection was performed by generating a transfection mix containing Env (HXB2/JR2/LASV): pR9ΔEnvΔNef: pcRev: Vpr (GFP/RFP): SER2/5 ratio of 2.68: 3.56: 1.2: 0.3: 0.3 μg, respectively, for a total plasmid quantity of 8 μg (for 10 cm dishes) or 2 μg (same plasmid ratio divided by a factor of 4) for 6-well plates. In the case of control viruses lacking SER5/2, an identical quantity of empty vector pCDNA3.1 was used. For TMEM16F containing virions, we used a mix of 0.1 μg of mTMEM16F and 0.2 μg of pCDNA 3.1 for 10 cm dishes, and 0.025 μg of mTMEM16F and 0.5 μg of pCDNA 3.1 for each well of 6-well plates, respectively. Post-transfection, the cells were incubated alongside the transfection mix for 8–12 h before exchanging the media with high-glucose DMEM (with P/S and 10% FBS) lacking phenol red. For immature virus preparations, transfections were carried out under identical conditions in the presence of 300 nM of Saquinavir to inhibit HIV-1 protease activity. A total of 48 h post-transfection, the supernatants were collected, passed through 0.45 μm PES filters to remove cell debris, and concentrated with Lenti-X concentrator prior to being aliquoted and stored at −80 °C.

The p24 content of viral preparations was determined by an ELISA-based method, previously described in [44,45]. For infectivity assays, TZM-bl cells were seeded in black–clear 96-well plates (Corning, NY, USA). Equivalent volumes of viruses across different viral preps were then added to the cells in the presence of the medium, supplemented with 20 mM of HEPES buffer. Then, the plates were centrifuged at 4 °C for 30 min at 1500× *g*. After centrifugation, the infected cells were incubated at 37 °C with 5% CO_2_ for 48 h before the addition of Bright-Glo luciferase substrate and subsequent incubation for 10 min at room temperature. The luminescence intensity of the samples was then immediately measured with a TopCount NXT reader (PerkinElmer Life Sciences, Waltham, MA, USA). The results were scaled to the respective p24 content of each prep.

Pseudovirus samples or whole-cell lysates of the producer cells were analyzed by Western blotting by loading ~0.2 ng of p24 for viral lysates or 6 ng total protein cell lysate onto 4–15% polyacrylamide gels (Bio-Rad, Hercules, CA, USA). The gels were then transferred onto 0.45 μm of nitrocellulose membrane (Cytiva, Marlborough, MA, USA) and then blocked with a 10% solution of dry milk suspended in 0.1% Tween 20 in PBS. After blocking, the membranes were subsequently kept overnight at 4 °C in the presence of primary antibodies prepared in 5% dry milk/0.1%Tween20/PBS under gentle agitation. The membranes were washed three times using 0.1% Tween 20 in PBS, prior to secondary antibody addition. Finally, the membranes were subsequently rinsed three times and then developed using chemiluminescence measurements on ChemiDoc XRS+ (Bio-Rad). The images were captured and analyzed using Image Lab V5.2 software. The following antibodies were used for Western blotting: primary antibodies—HIV IG (1:2000 dilution) and mouse anti-GAPDH (1:2000 dilution); secondary antibodies—goat anti-human HRP (1:3000 dilution) and rabbit anti-mouse HRP (1:3000 dilution).

### 2.3. Large Unilamellar Vesicle Preparation and Surface Immobilization

Large unilamellar vesicles (LUVs) of different compositions were made to test the specificity of Annexin V binding. The vesicles were prepared by utilizing extrusion techniques described elsewhere [27]. Briefly, lipid mixtures containing SM:Chol:DOPS or SM:Chol:DOPC (60 mol% SM:30 mol% Chol and 10 mol% DOPS or DOPC) were dissolved in chloroform and dried under a gentle stream of nitrogen/argon gas inside glass vials. The lipid mixtures also contained 1mol% Biotin-PE (for surface-immobilization) and 0.3 mol% NBD-PE (for fiducial signal). The dry lipid cake was stored under vacuum for at least two hours prior to hydration with 1× Annexin binding buffer. Following hydration, the solution was subjected to at least 5 freeze–thaw cycles before extrusion through a polycarbonate membrane (pore size 100 nm; Whatman/GE Healthcare) at least 20 times using an Avanti mini-extruder (Avanti Polar Lipids).

The protocol for immobilization of liposomes is reported elsewhere [27,46]. Briefly, a coverglass surface was coated with a mixture of fatty-acid-free Bovine Serum Albumin (BSA) and biotin-conjugated BSA at a molar ratio of 10:1. The surfaces were subsequently rinsed thoroughly with 1× Annexin binding buffer prior to treatment with 0.025 mg/mL of streptavidin solution for 30 min. Finally, a 2–4 μM suspension of liposomes was added, incubated for 10 min, and rinsed with buffer to create a single layer of immobilized liposomes for imaging.

### 2.4. Fluorescence Imaging of Surface-Immobilized Pseudoviruses and Lipid Vesicles

Immobilized pseudoviruses and lipid vesicles were imaged using a Zeiss LSM 880 microscope (Carl Zeiss Microimaging, Jena, Germany). After immobilization of vesicles (using BSA-Biotin) or pseudoviruses on 8-well chamber slides coated with poly-L-lysine (PLL) or poly-D-lysine (PDL), 5 µL of Annexin V–Alexa Fluor 647(AnxV-A647) stock was added to each well at a ratio of 1:40. The surfaces were incubated for at least 30 min at room temperature prior to imaging. All AnxV staining was performed in the presence of Annexin binding buffer unless otherwise specified. For viral permeabilization measurements, a similar protocol was followed. Prior to AnxV staining, the surface-bound virions were first treated with 100 units/well of pre-activated (treatment with 10 mM of DTT at 37 °C for 30 min) Streptolysin O (SLO) for 10 min at room temperature. The samples were then rinsed 4 times with AnxV binding buffer prior to AnxV staining.

For lipid order measurements, fresh Laurdan stock solutions were prepared at a concentration of 1 mg/mL in DMSO at 42 °C. Fully thawed stocks of pseudoviruses were first diluted (0.1–0.2 ng p24 in 50 µL of buffer) and then mixed with 0.5 µL of the 1 mg/mL Laurdan stock under continuous vortexing. All Laurdan stocks were then aliquoted, purged with Argon, and subsequently stored at −20 °C. Each vial was discarded after each thaw cycle. The mixture of pseudoviruses and Laurdan was incubated at 37 °C for 10 min and then subjected to centrifugation at 13,000 rpm for 5 min. Subsequently, the mixture was diluted at least 20-fold and bound to a poly-lysine-coated cover glass prior to imaging.

For immunofluorescence and co-staining experiments, iHA-containing viruses (SERINC5-iHA or SERINC2-iHA) were first immobilized in poly-lysine-coated 8-well coverglass chambers at room temperature. The samples were then fixed with 2% Paraformaldehyde (PFA) for 20 min prior to the addition of 20 mM of TRIS buffer to quench excess PFA. Following a buffer rinse, the surface was blocked by the addition of PBS++ supplemented with 15% FBS for at least 1 h at room temperature. The samples were then subjected to primary anti-HA antibody binding at room temperature for 2 h. Next, secondary staining was conducted using anti-mouse Alexa 568 at room temperature for 1 h. All the antibodies were prepared in PBS++ supplemented with 15% FBS. The samples were then rinsed at least four times with Annexin binding buffer prior to the addition of 5 µL of AnxV-A647 to each well at a 1:40 volume ratio. The samples were then incubated for at least 30 min before imaging.

### 2.5. Image Analysis for AnxV Quantification, Lipid Order, and Immunofluorescence Correlation

Single virus and liposome imaging was performed on a Carl Zeiss LSM 880 microscope (Carl Zeiss Microimaging, Jena, Germany) with a C-Apochromat 40×/1.2 W objective. Viral core markers GFP and RFP were excited by using 488 and 561 nm lasers, respectively. For AnxV staining experiments, both intraviral GFP-Vpr and mRFP-Vpr were used as fiducial signals and 647 nm was used for Annexin V–Alexa Fluor 647. The mRFP-Vpr viruses were used exclusively for Laurdan fluorescence imaging to prevent spectral overlap with GFP. For immunofluorescence experiments, GFP-Vpr viruses were used, which enabled us to use Alexa Fluor 568 conjugated antibody, and Annexin V–Alexa Fluor 647 simultaneously.

While both immunofluorescence and AnxV signal were collected in normal mode, Laurdan fluorescence was collected in lambda mode utilizing a wide emission window between 411 nm to 553 nm in 8.9 nm intervals. Lipid order was computed by methods described previously [27,47]. Briefly, the mRFP-Vpr signal was utilized to pinpoint the exact coordinates of the immobilized viruses (using local maxima detected using an algorithm adapted from fastpeakfind by Adi Natan https://www.mathworks.com/matlabcentral/fileexchange/37388-fast-2d-peak-finder (accessed on 7 March 2024)). By creating a binary mask utilizing the puncta exhibiting the mRFP signal, the Laurdan intensity at two different emission windows (443 nm and 513 nm) was extracted, and the respective generalized polarization (GP) values were calculated on a per particle basis using the following formula:Laurdan GP = (Intensity_443nm_ − Intensity_513nm_) + (Intensity_443nm_ + Intensity_513nm_)

Analysis of immunofluorescence and AnxV staining was performed in a similar fashion. For AnxV staining experiments, the emission data from all GFP/RFP-positive coordinates were extracted and analyzed. For AnxV co-staining experiments alongside iHA immunofluorescence, we extracted fluorescence intensities of iHA and AnxV channels from all GFP-Vpr-positive puncta using a wavelet-based object detection algorithm. iHA and AnxV puncta with signal intensities comparable to background levels and the ones containing saturated intensity pixels were discarded and the remainder of the intensity data were used for the correlation scatterplot. In all the cases, the puncta representing the fiducial signal from viruses (GFP, RFP) and liposomes (NBD-PE, Cy5-PE) were detected using a wavelet-based localization algorithm by open-source ICY software (version 2.5.2.0 accessed on 10 March 2024) [48].

### 2.6. Effects of Methyl-α-Cyclodextrin Treatment of Pseudoviruses

For testing the effects of viral lipid manipulation by MαCD, we adopted a slightly modified version of the luciferase-based luminescence assay. We first mixed ~0.2 ng p24 of viruses and either free MαCD or loaded with (10 mol%) DOPC/DOPS/DOPE/GM1/Biotin-PE (MαCD-X). For the concentration titration measurements, the virus quantity was held constant but the concentration of unloaded MαCD or the complex MαCD-X was varied (0 mM, 0.25 mM, 0.5 mM, and 1 mM MαCD-X, and 1 mM empty MαCD as a control). The MαCD: lipid molar ratio was maintained at 10:1 across all samples. All the samples were then incubated for 1 h at 37 °C and then chilled on ice before Lenti-X addition. The mixture was then incubated on ice for at least 4 h before centrifugation at 13,000× *g*. The supernatant was discarded and the pellet was resuspended in cold PBS+/+ and used for subsequent infectivity measurements. We observed little to no loss in p24 content of these viruses post-MαCD treatment. Infectivity measurements of individual preparations were compared utilizing Student’s *t*-test for statistical significance. For the Annexin V pre-binding experiment in Appendix A, we first bound ~0.2 ng of the virus to PDL-coated 96-well plate chambers. After 30-min of incubation on ice, we washed the surface with Annexin binding buffer (1× and then added either 5 μL PBS+/+ (0× AnxV) or 1.25 μL, 2.5 μL, and 5 μL of AnxV647 (0.25× AnxV, 0.5× AnxV, and 1× AnxV, respectively). TZM-bl cells were then overlaid atop AnxV-bound viruses for infectivity measurements, similar to a protocol described previously [27].

### 2.7. Statistical Analysis

Statistical significance between single-particle intensities for each sample (N > 100 puncta, unless otherwise specified) was estimated by using the Kolmogorov–Smirnov test. For imaging data involving co-staining of iHA and AnxV, the two-axis scatterplot was analyzed for correlation using Pearson’s correlation analysis, and the coefficient values between individual preparations were plotted. For data involving multiple independent biological replicates, the mean or median values for all preparations were combined and compared using an unpaired two-tailed *t*-test. All these analyses were performed using GraphPad PRISM 9 (San Diego, CA, USA) for Windows.

## 3. Results

To reliably test the effects of SER5-mediated enhancement in detectable external PS on HIV-1 particles, we utilized an imaging-based assay that measures fluorescence signal from single pseudoviruses immobilized on a microscope chamber surface. For the detection of PS on surface-immobilized viruses, we utilized AnxV, a vascular anticoagulant protein that is known to tightly bind to PS in a Ca^2+^-dependent fashion [49,50]. While this probe has been extensively used for the detection of apoptotic cells in a biological context, quantitative imaging of AnxV binding to pseudoviruses remains limited [28,51].

### 3.1. Quantitative Analyses of PS Signal on Single Pseudoviruses Demonstrates A Specific Increase in PS Signal upon SER5 Incorporation

To determine the utility of AnxV as a quantitative probe for PS on a single-particle level, we first tested the probe on surface-immobilized liposomes. Large unilamellar vesicles (LUVs) are a well-suited model system to mimic the diffraction-limited size (~100 nm) and characteristics (e.g., lipid composition) of viruses [27]. Two different liposome compositions, with and without PS, were tested: (1) SM/Chol/DOPS and (2) SM/Chol/DOPC. For both liposome formulations, we used 1 mol% of Biotin-PE, which aids in the near-irreversible immobilization of intact liposomes on a streptavidin-laden surface [52]. Additionally, we used 0.3 mol% of NBD-PE as a fiducial fluorescence marker, which helped us pinpoint the exact coordinates of the liposomes in the imaging field of view (FOV). The robust and specific AnxV signal detected on single PS-containing liposomes (see Section 2 and Appendix A) supports the utility of our approach for the quantification of PS at a single particle level.

Next, we sought to quantify the external PS content on surface-immobilized HIV-1 pseudoviruses containing the laboratory-adapted HXB2 Env, which is restricted by SER5 [19,53]. For AnxV staining experiments, pseudoviruses were labeled with either GFP-Vpr or mRFP-Vpr. To probe the effects of SER5 on PS exposure and virus infectivity, a panel of pseudoviruses was prepared that included control (Ctrl) and SER5-containing pseudoviruses, as well as viruses containing SER2, which lacks antiviral activity [10] and, therefore, serves as a negative control.

Unsurprisingly, functional characterization of viral preparations shows that SER5 incorporation leads to ~5–15-fold restriction relative to Ctrl, whereas SER2 incorporation was without effect (Figure 1A,B). To ensure robust incorporation of SER5 and SER2 into the virions, we performed immunofluorescence (IF) measurements by utilizing the HA tag inserted into the extracellular loop 4 of SER5 and SER2 (referred to as iHA), as previously described [24]. Staining the surface-immobilized virions with anti-HA Alexa Fluor 568 antibody (see Section 2) revealed robust staining of both SER5 and SER2 viruses, with SER2 exhibiting higher intensities relative to SER5 ([24], Figure 1C).

The GFP-Vpr-associated AnxV intensities of different viral preparations were quantified using imaging and image analysis protocols similar to those used for liposomes (Figure 1D–F). Despite a heterogenous distribution of AnxV intensities of single virions (Figure 1E), SER5-containing viruses had a significantly higher median AnxV intensity (~2 fold) relative to Ctrl and SER2 samples (Figure 1F). This trend was reproducible across multiple (N ~ 6) independent viral preparations. Our data are consistent with the report by Leonhardt et al. that suggests enhanced PS exposure on SER5- but not SER2-containing virions [28].

### 3.2. The External PS Signal Correlates with SER5 Incorporation into Virions

Given that levels of incorporation of both SER5 and SER2 into the viral lipid envelopes tend to vary [18,24] (Figure 1C), it was important to directly compare the relationship between the protein incorporation level and PS exposure. We, therefore, sought to determine if the incorporation levels of SER5 correlated with the PS signal on the viral surface.

Co-staining the viruses with the anti-HA 568 antibody and AnxV allowed us to accurately locate particles positive for SER2 or SER5 and exposed PS, respectively. By selecting GFP-Vpr puncta utilizing a wavelet-based detection algorithm (detailed in Section 2), we determined per-particle intensities of SER2 or SER5 and AnxV (see Figure 2C). We then performed (Pearson’s) correlation matrix analysis, wherein the wide range of iHA and AnxV intensities obtained from imaging SER5 and SER2 viruses from each preparation was tested for correlation (Figure 2C,D, [54]). By performing this analysis for three independent biological replicates, we were able to obtain an average Pearson’s coefficient (r) for both SER5 and SER2 samples: ~0.58 vs. ~0.23, respectively. This indicates that SER5 incorporation correlates better with enhanced PS exposure, both with regard to the overall median intensity values of independent preparations (Figure 1F) and on a per-particle basis (Figure 2C,D).

### 3.3. SER5 Incorporation Disrupts the Asymmetric Trans-Membrane PS Distribution

SER5 incorporation and PS signal are clearly correlated, but our approach does not differentiate between SER5-mediated externalization of PS and an increase in the overall viral PS content. While it has been shown that the overall lipid composition of viruses produced by SER5-expressing cells does not change, an imaging-based assay to directly stain PS on the inner leaflet can define the trans-membrane PS distribution in viral preparations tested. To that end, Streptolysin O (SLO), a pore-forming toxin belonging to the cholesterol-dependent cytolysin family, was used to permeabilize the viral membrane [55,56]. Given that SLO makes pores around 30 nm in diameter [56,57], AnxV should be able to permeate the viral lipid envelope, thereby allowing access to PS in the inner leaflet (Figure 3A).

Strikingly, a marked increase was observed in PS signal across all samples (Ctrl, SER5, and SER2) upon SLO permeabilization. By comparing AnxV intensities before and after SLO treatment, we were able to estimate the % of PS in the external leaflet (pre-permeabilization) relative to both the inner and outer leaflets (post-permeabilization). The % of PS in the external leaflet was consistently found to be higher for SER5 (~64%) relative to Ctrl and SER2 samples (~20%) (Figure 3B,C). These results led us to conclude that the enhancement in PS signal, indeed, is a result of the equilibration of the PS gradient across the viral membrane driven by SER5 incorporation. Of note, permeabilized SER5 virions consistently exhibited lower AnxV signal compared to control viruses and SER2, but not TMEM16F viruses (Figure 3B,C and Appendix A).

### 3.4. Enhanced PS Exposure on SER5 Virions Does Not Correlate Strongly with Specific Infectivity

To test if PS externalization indeed is an important feature of SER5’s restriction mechanism, we asked if the extent of PS exposure correlates with infectivity.

First, we sought to test if HIV-1 Env strains that are resistant to SER5 restriction (e.g., JR2 strain [17,53]) can maintain specific infectivity despite increased PS exposure mediated by SER5. We also tested pseudoviruses carrying the glycoprotein complex of unrelated Lassa virus (LASV-GPc) [58,59]. The infectivity and PS exposure were measured for HXB2, JR2, and LASV pseudoviruses containing SER5 or SER2 as a negative control. As expected, the incorporation of SER5 did not noticeably affect the infectivity of these pseudoviruses (Figure 4C). However, PS exposure was enhanced in both JR2 and LASV-GPc preparations containing SER5 samples relative to control samples (Figure 4A).

Second, we wanted to test the effects of incorporating a non-specific scramblase TMEM16F into HIV-1 pseudoviruses on infectivity and PS exposure. A constitutively active mTMEM16F D430G-FLAG mutant (for simplicity, abbreviated TMEM16F), which externalizes PS in the plasma membrane in a calcium-independent manner, was used [28,60]. This mutant carries an additional 21 amino acid sequence in its N-terminus and a D430G mutation, which improves incorporation into virions. While TMEM16F is known to promote PS externalization in both cells [60] and budding viruses [61], the protein is not widely regarded as a restriction factor, though Leonhardt et al. showed a modest reduction in infectivity of viruses containing this mutant.

While TMEM16F incorporation did result in particles with comparable PS exposure to SER5, the functional impact of incorporating TMEM16F was minimal. For comparable PS externalization levels (Figure 4B), SER5 is a much more potent restriction factor (Figure 4B,C) (~6.5× reduction in infectivity vs. Ctrl) than TMEM16F (~2.2× reduction), which is largely in agreement with prior observation [28]. Further characterization of these TMEM16F viruses revealed that the overall lipid order, reported by the Laurdan probe, is not much different from SER5 or SER2 viruses (Appendix A). Moreover, SLO permeabilization reveals that TMEM16F viruses exhibit ~50% external PS (Appendix A), implying that it behaves similarly to SER5 viruses with regard to its ability to disrupt viral PS asymmetry (Appendix A). Taken together, these results imply that PS externalization upon SER5 incorporation does not strongly correlate with a reduction in infectivity.

Finally, given that Gag-processing during HIV-1 maturation is critical for viral infectivity [62], we then asked if Gag-processing can play a role in SER5-mediated PS externalization. Given that HIV-1 maturation is known to affect viral lipid order [35], as well as lead to the removal of several inner-leaflet lipids [63] and affect membrane rigidity [64], we wanted to perform additional control experiments to see if immature Gag lattice can diminish the SER5-induced PS externalization. To test for this possibility, Ctrl and SER5-containing virions were produced in the presence of HIV-1 protease inhibitor, Saquinavir (SQV), resulting in immature particles [65,66]. While SER5 incorporation did not significantly affect the background-level infectivity of SQV samples (Appendix A), the PS exposure levels were enhanced overall by SER5 compared with immature Ctrl and SER2-containing samples (Appendix A).

### 3.5. SER5 Restricts Infectivity of Viruses Produced by Cells with Constitutively High Levels of Externalized PS

To understand if intrinsic lipid asymmetry of the producer cell plasma membrane plays a role in SER5 restriction, we turned our attention to manipulating the external PS content of virus-producing cells. We reasoned that, if PS externalization alone causes lower infectivity of SER5-containing viruses, then viruses produced by cells containing higher baseline levels of external PS should also exhibit high levels of external PS and, thus, no longer be susceptible to SER5 restriction. To that end, we utilized HEK293T-CDC50a KO cells (referred to as ΔCDC50a), which lack the gene coding for CDC50 [37,67]. This protein facilitates the folding and localization of plasma membrane flippase complexes, including Type 4 P-Type ATPase, which is responsible for maintaining PS asymmetry at the plasma membrane [68]. Cells lacking CDC50a have been shown to exhibit higher levels of outer leaflet PS under non-apoptotic conditions unlike WT HEK293T cells [37]. We reasoned that the viruses produced from these cells would feature higher levels of outer leaflet PS relative to those produced from WT HEK 293T cells [69].

We produced HIV-1 pseudoviruses in both WT and ΔCDC50a HEK293T cells in parallel to compare relative PS exposure and the functional impact of disrupting PS asymmetry at the producer cell level. Unsurprisingly, viruses produced from ΔCDC50a cells exhibited a higher baseline level of external PS, as evidenced by a high overall AnxV median intensity across multiple preparations (Figure 5A,B). Interestingly, SER5 viruses produced from these KO cells exhibit only a minor (~1.1 fold) increase in AnxV staining relative to the Ctrl (Figure 5A,B). This is in stark contrast with virions produced from HEK293Ts, where a ~2-fold increase in AnxV intensity was observed for SER5 viruses relative to the Ctrl (Figure 1D and Figure 5B). This result implies that the virus membrane asymmetry is primarily defined by the intrinsic asymmetry of the plasma membrane at the budding site. Given previous reports of SER5 incorporation not leading to changes in overall PS levels [26], it is likely that SER5 carries out a scramblase-like function, similar to TMEM16F, leading to loss of lipid asymmetry.

To further test the correlation between SERINC incorporation and PS exposure on viruses produced from ΔCDC50a cells, we stained the viruses using AnxV, while simultaneously immuno-staining against iHA, like in Figure 2C. We observed no clear correlation (r ~ 0.38) between SER5 levels and external PS in viruses produced from these KO cells (Figure 5E), unlike the viruses produced from WT cells, where we observed a highly significant correlation (Figure 5D). Upon permeabilization of these viruses with SLO, like in Figure 3, we observed ~54, 76, and 68% external PS in the Ctrl, SER5, and TMEM16F samples, respectively (Appendix A). These observations clearly indicate that the overall increase in AnxV staining observed in ΔCDC50a cell-produced viruses is mostly a result of disruption to the producer cell’s flippase activity, without the need for the scramblase-like activity of SER5.

Finally, we wanted to test the functional impact of virus production in ΔCDC50a cells. The infectivity measurements reveal a marked decrease in the overall specific infectivity of virions produced from ΔCDC50a relative to the WT-HEK293 T-produced samples (Figure 5F). We performed additional characterization of these viruses using Western blotting and observed no significant impact on Gag processing or viral protein expression relative to viruses produced from WT 293T (Appendix A). We ascribe this loss in infectivity to potential pleiotropic effects arising from CDC50a deletion, which might cause compensatory disbalance for many factors besides just PS asymmetry. However, we cannot fully eliminate the potential role of PS in the overall infectivity reduction observed in these viruses relative to WT-HEK293 T-produced viruses. Importantly, the SER5 restriction phenotype (~6-fold reduction in infectivity vs. Ctrl samples) becomes apparent after the normalization of infectivity data to the respective Ctrl viruses and is well conserved regardless of the producer cells used (Figure 5F,G). This further supports our earlier observation that the enhanced PS staining does not appear to be directly related to the SER5’s restriction mechanism.

### 3.6. Manipulation of Viral Lipid Composition Reveals That SER5 Restriction Is Unrelated to External PS Content of the Viruses

Having gained mechanistic insights into SER5-mediated HIV-1 restriction from endogenous lipid manipulation of producer cells, we wanted to directly test the effects of modulating viral lipids exogenously. This has been performed in the context of cholesterol exchange in the past, but the literature on exogenous viral lipid exchange outside cholesterol remains very sparse [70]. Ward et al. [25] found that PE incorporation, and not PS incorporation, led to a rescue of fusion of SER5-containing viruses, consistent with our data showing a lack of correlation between PS exposure and infectivity. To reliably incorporate exogenous lipids into pseudoviruses, without compromising their functional integrity, we utilized methyl-α-cyclodextrin (MαCD) that facilitates lipid exchange (Figure 6A) [71,72]. MαCD has been utilized for lipid exchange at a plasma membrane level and for in vitro applications for many years [73]. However, to the best of our knowledge, no prior study has characterized MαCD-mediated lipid exchange with the viral lipid envelope.

To find optimal conditions for MαCD-mediated lipid exchange, we first treated viruses with empty, unloaded MαCD at different concentrations (0, 0.25, 0.5, and 1 mM). Even at the highest concentration tested (1 mM), the functional impact of MαCD treatment on these viruses was negligible (Figure 6B and Appendix A).

To test the functional effects of specific lipid exchange at the viral lipid envelope, we prepared three independent MαCD complexes (MαCD-X) comprising one of the three lipids denoted by X: (1) DOPS, (2) DOPC, and (3) DOPE. The virus samples were treated with 0, 0.25, 0.5, and 1 mM of the MαCD-X complex. In all lipid-exchange experiments, 1 mM of unloaded MαCD was included as a control. Treatment with MαCD-DOPS led to a nearly 200% increase in the infectivity of SER5-containing viruses (in other words ~2.5–3-fold reduction in restriction potency of SER5), while no effect was detected in control viruses (Figure 6B,C,E and Appendix A).

Pretreatment with 1 mM of MαCD alone decreased the AnxV binding to single virions by 18–42% relative to the original intensity values of the respective untreated control across all samples (Ctrl, SER5, SER2, and TMEM16F) (Figure 6D,E). This indicates that even empty MαCD can non-specifically remove PS (and likely other lipids) from the surface of virions, leading to weaker AnxV staining, without having a major impact on infectivity. Unfortunately, direct imaging of MαCD-DOPS-treated viruses was not possible due to strong background signals as a result of treating the viruses with the lipid complex. No statistically significant changes to infectivity were observed in the case of DOPE/DOPC, indicating that the mild recovery of viral infectivity is specific to PS exchange. It should be stressed that exogenous PS exchange selectively augments but does not decrease the infectivity of SER5 pseudoviruses, as would be expected if elevated PS levels in the external leaflet inhibited viral fusion.

Additionally, we tested the functional impact of blocking the accessible external PS on surface-bound pseudoviruses with AnxV 647. Viruses were adhered to the bottom of 96-well plates, pretreated with AnxV 647, and overlaid with cells (see Section 2). We observed a modest (up to ~2-fold) reduction in infectivity with increasing AnxV concentrations, which is largely consistent with the previous study that employed a similar assay by treating HIV-1 particles with AnxV in solution [38]. However, no significant differences were observed between Ctrl, SER5, and SER2 across the concentration range tested (Appendix A). The modestly decreased virus infectivity upon PS sequestration further suggests that external PS is a positive regulator of HIV-1 entry/fusion.

## 4. Discussion

Here, we investigated the role of PS externalization in SER5-mediated restriction of HIV-1. A single virus-based analysis of AnxV binding revealed that SER5 incorporation is associated with a higher level of external viral PS, which is consistent with the previous report [28]. Notably, the PS signal in the external leaflet of the viral membrane correlates with the amount of virus incorporated in SER5 but not SER2.

However, further investigation revealed that the elevated level of PS on the virus surface does not always correlate with reduced infectivity. Firstly, whereas SER5 incorporation into HIV-1 particles elevates the level of external PS, the ability of SER5-resistant viral glycoproteins (HIV-1 JR2 Env or LASV GPc) to mediate virus–cell fusion is not significantly affected. Secondly, in agreement with the previous study [28], the incorporation of TMEM16F scramblase increases the PS levels, similar to those on SER5-containing virions, but does not reduce HIV-1 infectivity nearly as potently as SER5 incorporation. Thirdly, HIV-1 infectivity significantly increased under conditions conducive to MαCD-mediated delivery of exogenous DOPS, but not DOPE or DOPC, into the membrane of SER5-containing pseudoviruses. Fourthly, SER5 incorporation still reduces the infectivity of HIV-1 pseudoviruses exhibiting elevated external PS levels due to disruption of lipid asymmetry in the plasma membrane of producing cells.

Importantly, we developed an assay to assess the PS asymmetry across the viral membrane, which combines AnxV staining with membrane permeabilization with SLO that ensures access of this PS probe to the virus interior. The difference between the AnxV signal from intact vs. permeabilized viruses reflects trans-membrane PS asymmetry, with only a small fraction of PS present in the external leaflet of control or SER2 pseudoviruses. This asymmetry is completely lost upon incorporation of SER5 or TMEM16F. A slight decrease in total PS levels in SER5-containing viruses relative to Ctrl and SER2 post-permeabilization was observed (Figure 3B,C). While the reason behind this reduced intensity is not clear, it is possible that AnxV binding to SER5 virions can be modulated by the probe’s sensitivity to membrane curvature [74]. An alternative possibility is that SER5 sequesters a portion of PS making it inaccessible for AnxV. Importantly, we have a clear indication from permeabilized TMEM16F viruses (Appendix A) that SER5 does indeed behave like a scramblase. These important findings, along with the elevated exposure of PS on virions produced by ΔCDC50a cells, support the notion that the plasma membrane lipid (PS) asymmetry is largely maintained in HIV-1 virions and that SER5 nearly equilibrates the PS content of inner and outer leaflets of the viral membrane. Given that the cross-bilayer flip-flop rates of viral lipids are very slow, it is likely that the overall viral lipid (PS) asymmetry (or the lack thereof, in the presence of a scramblase) is dictated at the plasma membrane of the budding site of the virus-producing cell.

Unsurprisingly, SER5 viruses produced from ΔCDC50a cells exhibit only a very minor increase in PS staining relative to Ctrl and SER2 (in stark contrast with viruses produced from HEK293T). However, at the same time, SER5 exhibits almost identical restriction potency for viruses produced from both HEK293T and ΔCDC50a cells. If the scramblase activity of SER5 was an important aspect of its restriction mechanism, the lack of PS asymmetry at the producer cell level should have a strong effect on the antiviral activity of SER5. This finding, therefore, demonstrates that PS externalization is likely not the key driver for infectivity reduction by SER5.

The ΔCDC50a cell-produced pseudoviruses were generally less infective than control viruses. While the pleiotropic effects of deleting a key regulatory protein like CDC50a are likely responsible for the significant functional impact on the viruses, independent validation was required to test the exact role of external PS on viral infectivity. To that end, pretreatment with MαCD causes an overall reduction in PS levels on the viruses (18–40%) without a significant reduction in infectivity in the Ctrl, SER5, and SER2. On the other hand, pretreatment with a MαCD-DOPS complex antagonizes SER5 restriction, an effect was not observed upon treatment with MαCD-DOPE or MαCD-DOPC.

It is not clear if the MαCD-DOPS actively antagonizes SER5 due to PS enrichment on the viral membrane or due to other off-target effects from MαCD-mediated lipid exchange. Direct imaging of viral PS after MαCD-DOPS treatment was not possible due to severe background issues. However, we believe that exogenous manipulation of viral lipids by MαCD, especially the absence of functional consequences from the overall reduction in PS signal, further supports the lack of correlation between PS exposure and infectivity. Taken together, the purported disruption of PS asymmetry (and, likely, the overall lipid asymmetry) of the viral membrane by SER5 does not form the basis for its restriction mechanism. Instead, we speculate that a more complex mechanism is likely responsible for the antiviral activity of SER5.

The exact mechanism of SER5 restriction and its link to lipid composition or asymmetry is not clear. One possibility is that SER5 incorporation can lead to the disruption of lipid nano-domains on the viral lipid envelope, as proposed previously [25]. Although, we have shown that the global lipid order does not correlate strongly with SER5 restriction, ultrastructural data from cryo-EM indicate that a nanoscale-level heterogeneity of the viral membrane might play a role in the restriction mechanism [25]. It is also possible that SER5 disrupts other viral lipids besides PS, leading to other downstream effects on the fusogenicity of these viruses. The putative lipid-binding pocket in SER5 can bind cholesterol, cardiolipin, and other lipids [75]. Lipids, such as cardiolipin, PIP2, and SM have been implicated in HIV-1 fusion [76,77,78] and infection, and while SER5 does not appear to affect the overall lipid composition of virions [26], scrambling or redistribution of these lipids can have deleterious effects on viral fusion, leading to infectivity reduction. Indeed, it is known that SER5 can modulate Env conformation and predispose them to premature inactivation, and a lipid-mediated, indirect mechanism, as discussed above, is plausible [21,53]. More experiments are needed to comprehensively test these possibilities. By adding some key insights to our current understanding of the relationship between SER5 and viral lipid membrane, we believe that our study will pave the way for further quantitative investigation in this area.

## 5. Conclusions

Collectively, our results further validate the scramblase-like function of SER5 in HIV-1 membrane. However, since the external PS levels do not strictly correlate with SER5’s restriction potency, we conclude that PS externalization is not directly related to HIV-1 restriction which likely occurs via a more complex mechanism.

## Figures and Tables

**Figure 1 biomolecules-14-00570-f001:**
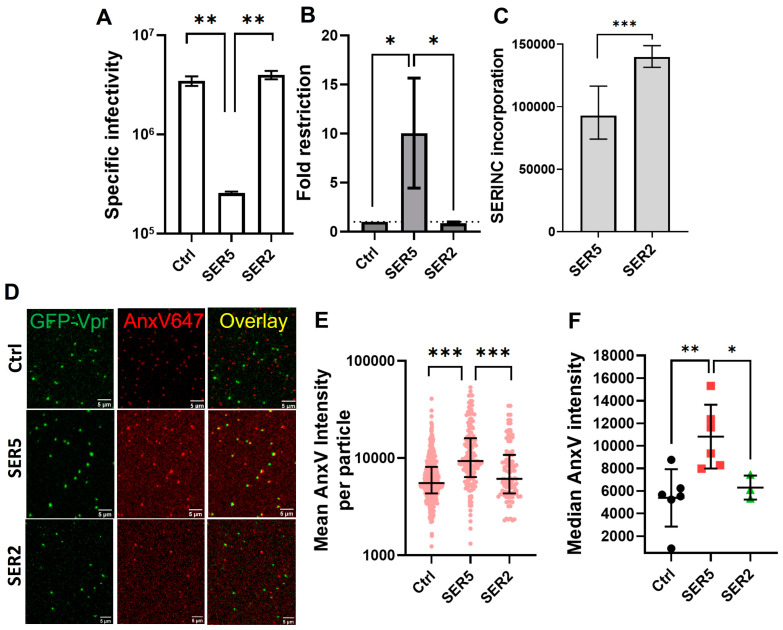
SER5 incorporation increases PS exposure on HIV-1 pseudoviruses. (**A**) Representative luciferase reporter-based single-round infectivity assay. Specific infectivity was plotted as the luciferase signal per ng of the p24 virus used for the assay (RLU/ng). (**B**) Fold infectivity reduction relative to Ctrl samples was plotted as the fold restriction from 4 independent biological replicates. (**C**) Representative immunofluorescence (IF) intensity from SER5 and SER2 samples showing overall levels of iHA incorporation. The bars and the error bars in (**A**–**C**) represent the mean and S.D., respectively. (**D**) Representative confocal images of Ctrl (**upper panel**), SER5 (**middle panel**), and SER2 (**lower panel**) pseudoviruses containing GFP-Vpr (green) and stained with AnxV A647 (red); scale bar ~ 5 µm. (**E**) Representative scatterplot showing mean AnxV intensities (in arbitrary units, AU) of single viral particles in each sample (N > 100). (**F**) Median single virus AnxV intensity values from different viral preparations are plotted; each symbol represents an independent biological replicate. The horizontal lines on both (**E**,**F**) represent the median values, while the error bars represent the interquartile range. Statistical analysis for (**E**) was performed by using the Kolmogorov–Smirnov test. (**A**–**C**,**F**) were analyzed using Student’s *t*-test. *, 0.05 > *p* > 0.01; **, 0.01 > *p* > 0.001; ***, *p* < 0.001.

**Figure 2 biomolecules-14-00570-f002:**
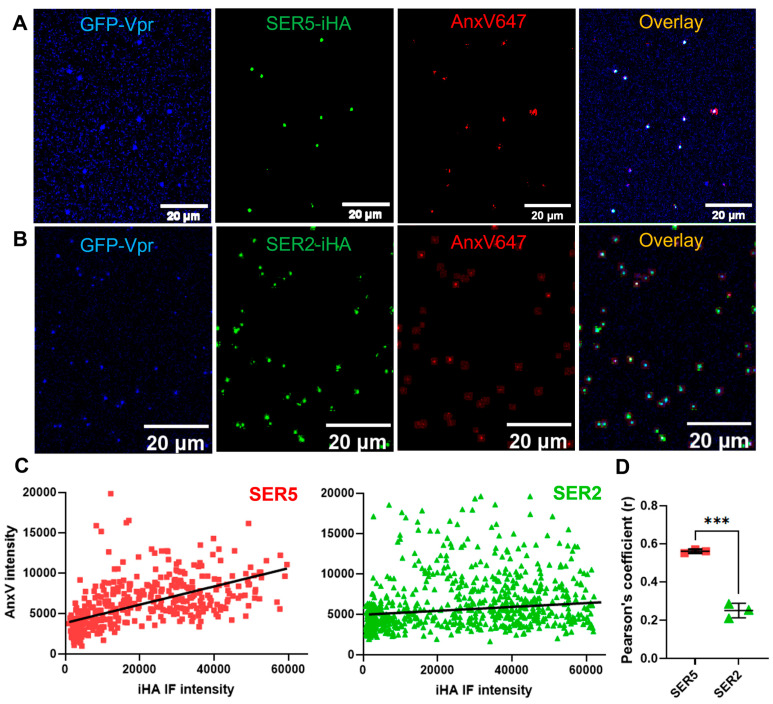
PS staining correlates with SER5 incorporation into virions. (**A**,**B**) Representative confocal images of SER5 (**A**) or SER2 (**B**) pseudoviruses containing GFP-Vpr (blue) and stained by both anti-HA CF568 antibody (green) and AnxV647 (red). For clarity, background-subtracted images are presented on the overlay panel. (**C**) Representative correlation scatterplot of AnxV intensities and iHA IF intensities of single viral particles (N > 100 each preparation). (**D**) Pearson’s coefficient values (symbols) were obtained from analyzing correlation scatterplots like (**C**) from three independent biological replicates. The horizontal lines in (**D**) indicate the mean, and symbols indicate values from each replicate and were analyzed using Student’s *t*-test. ***, *p* < 0.001.

**Figure 3 biomolecules-14-00570-f003:**
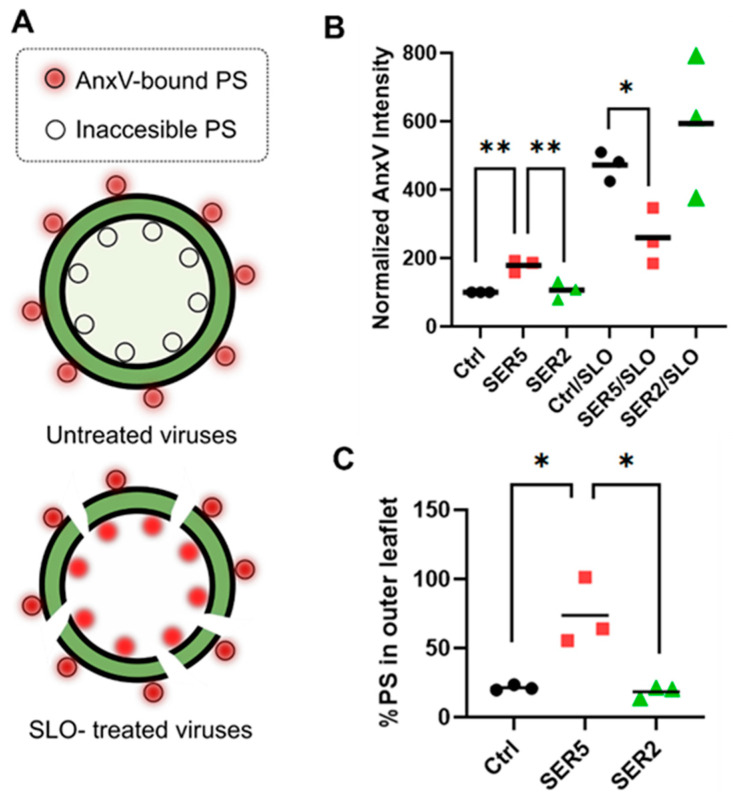
SER5-mediated disruption of trans-bilayer PS asymmetry revealed by viral membrane permeabilization. (**A**) A schematic diagram representing PS staining by AnxV before and after viral membrane permeabilization by SLO. (**B**) Normalized median AnxV intensity values (% of Ctrl) from 3 independent biological replicates. (**C**) The % of PS in the outer leaflet of the viruses from 3 independent biological replicates was calculated from data in B and plotted. The symbols indicate values from each replicate, and the horizontal line denotes the mean value. Statistical analysis was performed using Student’s *t*-test. *, 0.05 > *p* > 0.01; **, 0.01 > *p* > 0.001.

**Figure 4 biomolecules-14-00570-f004:**
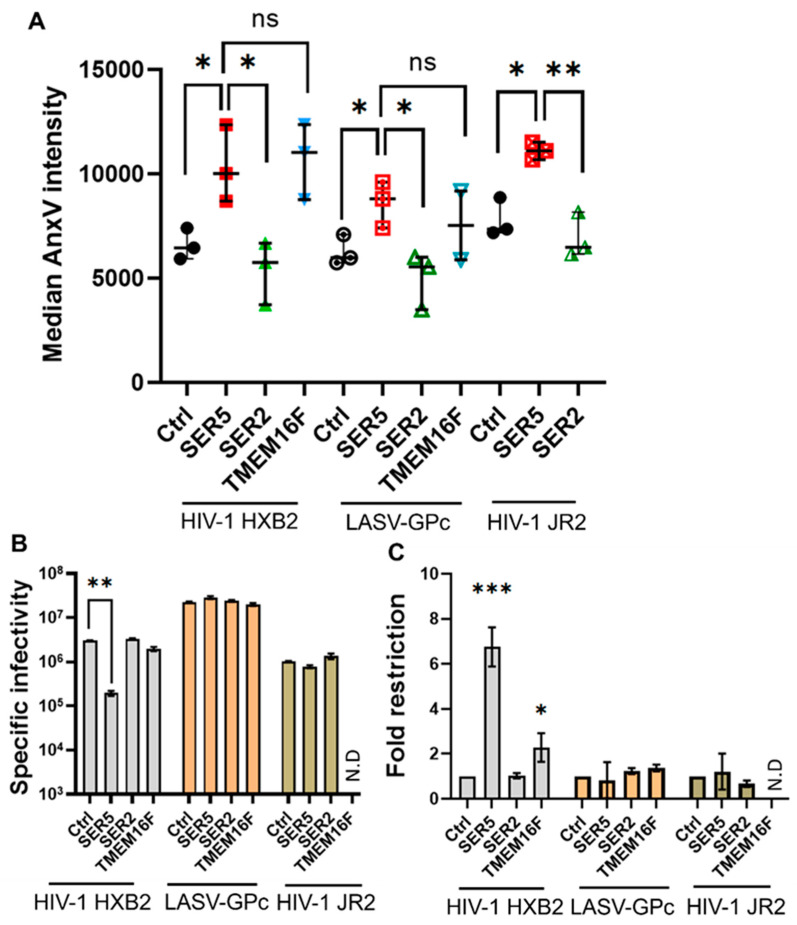
Enhancement in PS externalization does not correlate strongly with SER5-mediated restriction. (**A**) Median AnxV intensity values from 3 biological replicates are plotted and analyzed. Symbols denote values from each biological replicate. The middle horizontal lines and error bars represent mean and S.D., respectively. (**B**) Representative luciferase reporter assay showing the functional impact of using SER5-resistant LASV GPc (orange bars) and HIV-1 JR2 Env (green bars) compared to HIV-HXB2 (grey bars). Specific infectivity was plotted as the luciferase signal per ng of the p24 virus under each condition. (**C**) Fold infectivity reduction relative to the Ctrl was plotted as the fold restriction from 3 independent biological replicates. The bars and the error bars represent the mean value and S.D., respectively. Statistical analysis was performed using Student’s *t*-test. n.s., *p* > 0.05; *, 0.05 > *p* > 0.01; **, 0.01 > *p* > 0.001; ***, *p* < 0.001. Note that *p* > 0.05 is not shown in (**B**,**C**) to aid visual clarity. N.D, not determined.

**Figure 5 biomolecules-14-00570-f005:**
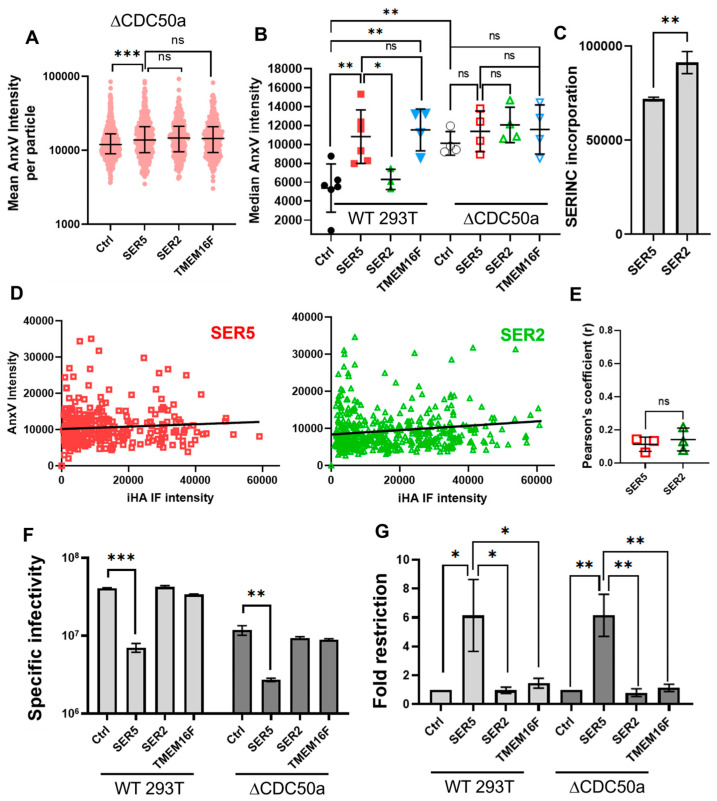
Pseudoviruses produced by ΔCDC50a cells are restricted by SER5 despite a higher baseline level of external PS. (**A**) Representative scatterplot showing mean AnxV intensities of single viral particles (N > 100). (**B**) Median AnxV intensity values from 4 independent pseudovirus preparations produced by ΔCDC50a cells are plotted. Each symbol represents an independent biological replicate. For ease of comparison, the median AnxV intensity distribution for viruses produced by WT 293T cells is replotted from Figure 1D. The horizontal lines on both (**A**,**B**) represent the median value, while the error bars represent the interquartile range. Symbols from (**B**) denote the median intensity value from each biological replicate. (**C**) Representative IF intensity from SER5 and SER2 samples showing overall levels of iHA incorporation. (**D**) Representative correlation scatterplot of AnxV intensities and iHA IF intensities from single viral particles produced from ΔCDC50a cells. (**E**) Pearson’s coefficient values (symbols) were obtained from analyzing correlation scatterplots from three independent biological replicates exemplified in (**D**). The horizontal line represents the mean and error bars of the S.D. (**F**) Representative luciferase reporter assay readings showing the difference in infectivity of viruses produced from WT and ΔCDC50a cells. Specific infectivity was plotted as the luciferase signal per ng of the p24 virus under each condition. (**G**) Fold infectivity reduction relative to the Ctrl samples was plotted as the fold restriction from 5 (WT 293T viruses) and 6 (ΔCDC50a viruses) independent biological replicates. The bars and the error bars represent the mean value and S.D., respectively. Statistical analysis was performed using Student’s *t*-test. n.s., *p* > 0.05; *, 0.05 > *p* > 0.01; **, 0.01 > *p* > 0.001; ***, *p* < 0.001. Note that *p* > 0.05 is not acknowledged in (**F**,**G**) to aid visual clarity.

**Figure 6 biomolecules-14-00570-f006:**
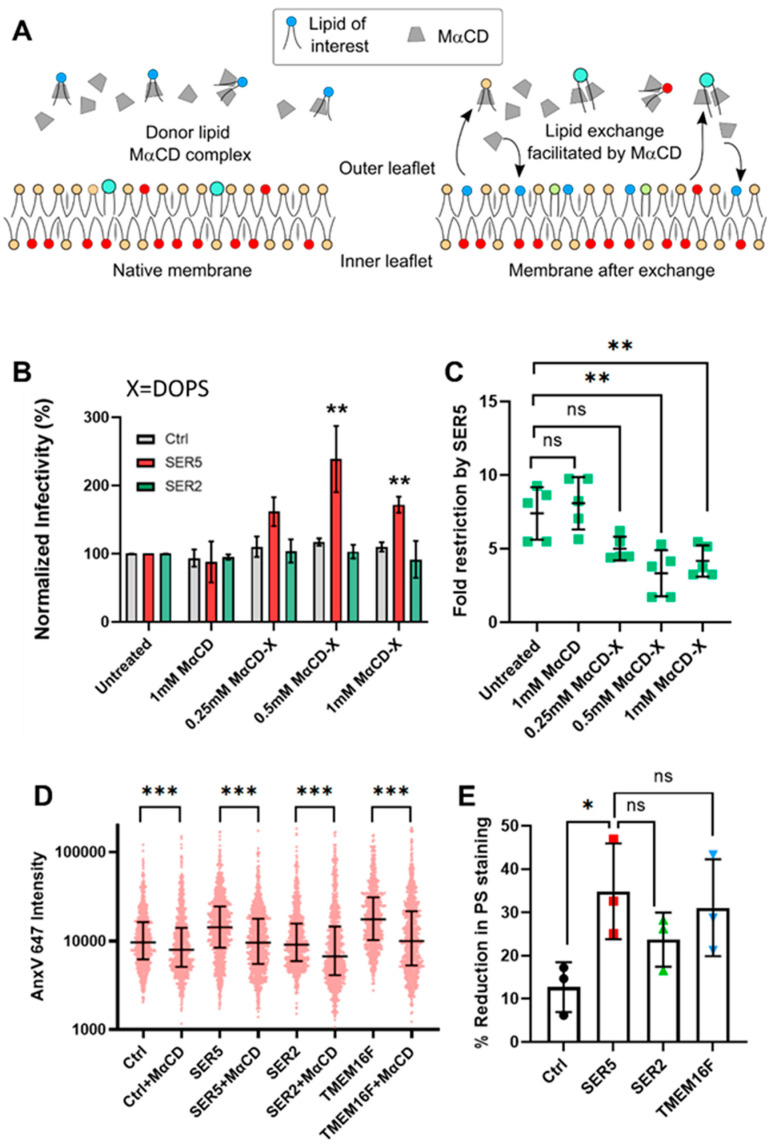
MαCD-mediated disruption of outer leaflet PS of HIV-1 pseudoviruses has little impact on SER5 restriction. (**A**) A schematic diagram representing MaCD-mediated lipid exchange with the membrane surface (adapted from [71]). (**B**) Functional effects of treating viruses with MaCD-DOPS complex. Infectivity data obtained from 5 independent biological replicates are normalized to the respective untreated samples and plotted. The bars and the error bars in (**C**,**D**) represent the mean value and S.D., respectively. (**C**) The effects of MaCD-DOPS treatment on restriction potency of SER5. Fold infectivity reduction by SER5 from five independent biological replicates (green symbols) is plotted. (**D**) Representative scatterplot showing mean AnxV intensity distribution before and after treatment with 1 mM of empty MaCD. The data are for N > 100 viral particles in each sample. The middle horizontal lines represent the median value, while the error bars represent the interquartile range. (**E**) Reduction in PS staining was obtained by comparing median AnxV intensity values before and after treatment with 1 mM of MaCD in panel (**B**). The horizontal line and error bars denote the mean value and S.D., respectively. Statistical analysis for (**D**) was performed using the Kolmogorov–Smirnov test. Statistical analysis on (**B**,**C**,**E**) was performed using Student’s *t*-test. n.s., *p* > 0.05; *, 0.05 > *p* > 0.01; **, 0.01 > *p* > 0.001; ***, *p* < 0.001. Note that *p* > 0.05 is not acknowledged in (**B**) to aid visual clarity.

## Data Availability

Data are contained within the article and Appendix A.

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
