# Peer review of "Disruption of Transmembrane Phosphatidylserine Asymmetry by HIV-1 Incorporated SERINC5 Is Not Responsible for Virus Restriction"

_biomolecules, 2024, doi:10.3390/biom14050570_

Round 1

Reviewer 1 Report

Comments and Suggestions for Authors

Raghunath and colleagues demonstrated that the inclusion of SERINC5 into HIV-1 particles is associated with elevated levels of PS. Interestingly, this correlation was not observed with SERINC2. Furthermore, through the use of SLO, the researchers showed that the increase in PS signal on the virus's exterior is a result of the equilibration of the PS gradient.

Despite artificially externalizing PS using a scramblase, HIV-1 infection was not significantly blocked or affected. It is noteworthy that the scramblase exhibited behavior similar to SERINC5, externalizing approximately 50% of PS. Manipulating PS levels on the virus's surface led the researchers to conclude that the heightened PS levels in viruses containing SERINC5 are not directly linked to restriction. This is an important and timely study.

The study's methodology is good and well-controlled. However, prior to publication, an important control is needed for Figure 5. Specifically, virus expression in producer cells and supernatants via Western blot would be crucial(anti-p24). This control is important to show whether deltaCDC50a cells have a defect in viral expression or viral release when compared to wild type cells. 

Author Response

“Raghunath and colleagues demonstrated that the inclusion of SERINC5 into HIV-1 particles is associated with elevated levels of PS. Interestingly, this correlation was not observed with SERINC2. Furthermore, through the use of SLO, the researchers showed that the increase in PS signal on the virus's exterior is a result of the equilibration of the PS gradient.

Despite artificially externalizing PS using a scramblase, HIV-1 infection was not significantly blocked or affected. It is noteworthy that the scramblase exhibited behavior similar to SERINC5, externalizing approximately 50% of PS. Manipulating PS levels on the virus's surface led the researchers to conclude that the heightened PS levels in viruses containing SERINC5 are not directly linked to restriction. This is an important and timely study.”

We thank the reviewer for acknowledging the importance of the study, and an accurate summary of the work we had presented in our manuscript.

“The study's methodology is good and well-controlled. However, prior to publication, an important control is needed for Figure 5. Specifically, virus expression in producer cells and supernatants via Western blot would be crucial(anti-p24). This control is important to show whether deltaCDC50a cells have a defect in viral expression or viral release when compared to wild type cells.”

We welcome the constructive feedback with regards to the lack of anti-p24 Western blots. We would like to point out that all our reported specific infectivity data (Figure 1A, 4B and 5G) and fold restriction data (Figure 1B, 4C and 5H) were all plotted after accounting for the slight overall differences in the p24 content of the respective viral preparations, measured by a p24 ELISA assay described in the manuscript.

However, we agree that additional validation of our viruses will be useful, especially given the unconventional nature of ΔCDC50a cells (used in some experiments. To that end, we produced 3 additional panel of viruses in ΔCDC50a cells (containing Ctrl, SER5, SER2 and TMEM16F) and 2 panels of independent biological replicates from viruses produced from WT 293T to identify any potential viral defects.

We have included representative Western blotting data in the supplemental figures (Figure S7). For the reviewer’s convenience, we are additionally attaching the same figure in this rebuttal document including the fold-restriction data.

While we do observe small changes in Gag-processing efficiency in one of our recent viral the overall viral expression, and Gag-processing efficiency appears comparable between viruses produced from WT and ΔCDC50a 293T cells. Furthermore, we were able to successfully reproduce the data we reported in the original version of the manuscript, with regards to SER5 largely retaining its restriction potency, despite the high baseline PS levels in viruses produced from ΔCDC50a cells. We have added the additional biological replicates to Figure 5G of the revised manuscript.  

Of note, we detect a minor fraction of processed Gag in cell lysates. Interestingly, we see a higher amount of processed Gag (p24) in cell lysates of 293T producer cells, as opposed to ΔCDC50a cells. This could be explained by many factors, including but not restricted to potential differences in transfection efficiency between the cell lines. However, we are confident, from our Western blot results and reproducible infectivity data, that viruses produced from these cells are not defective to any significant degree. They exhibit comparable Gag-processing albeit with reduced overall infectivity (acknowledged in the revised manuscript). Most importantly, the viral infectivity is strongly restricted by SER5 expressed in CDC50a cells, to a comparable extent to viruses produced from 293T cells, adding further validation to our initial conclusions. We have added some more text to provide additional context in the revised version of the manuscript to reflect the newly acquired data.

We once again thank the reviewer for the useful and constructive feedback.

Reviewer 2 Report

Comments and Suggestions for Authors

Restriction factors such as SERINC5/3 preventing HIV and host cell fusion through Env inactivation may involve additional mechanisms and even other cellular functions developed during evolution. Thus, the responsibility of SERINC5/3 for modulating the distribution of phosphatidylserine on the viral surface could be one such additional mechanism to support antiviral function. Using a number of complementary methods, the authors of this work have attempted to clarify this assumption.

The results of the work will be of interest to virologists, and the methodology of the experiment will be useful for graduate students, undergraduates and teachers of medical microbiology and virology.

Several good reviews have recently been published demonstrating mechanisms for antiviral functions against retroviruses. As a side note, I would like to recommend the authors to mention in the introduction one such important review: Shaofen Xu et al, The Emerging Role of the Serine Incorporator Protein Family in Regulating Viral Infection, Front. Cell Dev. Biol., (2022) 10:856468. doi: 10.3389/fcell.2022.856468.

Author Response

“Restriction factors such as SERINC5/3 preventing HIV and host cell fusion through Env inactivation may involve additional mechanisms and even other cellular functions developed during evolution. Thus, the responsibility of SERINC5/3 for modulating the distribution of phosphatidylserine on the viral surface could be one such additional mechanism to support antiviral function. Using a number of complementary methods, the authors of this work have attempted to clarify this assumption.

The results of the work will be of interest to virologists, and the methodology of the experiment will be useful for graduate students, undergraduates and teachers of medical microbiology and virology.

Several good reviews have recently been published demonstrating mechanisms for antiviral functions against retroviruses. As a side note, I would like to recommend the authors to mention in the introduction one such important review: Shaofen Xu et al, The Emerging Role of the Serine Incorporator Protein Family in Regulating Viral Infection, Front. Cell Dev. Biol., (2022) 10:856468. doi: 10.3389/fcell.2022.856468.”

We thank the reviewer for the kind and insightful comments. We agree that the manuscript could benefit from citing review articles that summarizes some great work in the field with respect to SERINC restriction. To that end, we have added a couple of citations to our manuscript, including the one recommended by the reviewer.

Reviewer 3 Report

Comments and Suggestions for Authors

Raghunath et al reported an interesting finding that disruption of PS distribution associates with incorporated SER5 protein HIV-1 envelope but does not determine restrictive function of SER5. This may help understand the complex mechanism of HIV host restriction and potential therapeutic applications. Several points have to be addressed to have it published.

Major points:

1. In Figure 2B-C and Figure 5 D-F, what’s the biological relevance of Pearson’s coefficient? Note that in SER2 treatment most virions have lower iHA intensity, which suggests less efficient incorporation of SER2 than SER5. This raise a question whether a linear model is appropriate, since it’s difficult to assess the relationship between protein incorporation and PS externalization if the protein is less or even not incorporated.

2. Note that in Figure 1C and Figure 5C, SER2 incorporation is significantly higher than SER5 as measured by iHA fluorescence intensity. These data are not in line with Figure 2B and Figure 5D. Please consider possible reasons.

3. Please include control fluorescent images for Figure 1 (Ctrl) and Figure 2 (SER2-iHA).

4. The fluorescence images are quantified as intensity which seems to have an arbitrary unit. Could author please describe how to make readings from different channels (annexin V and iHA) comparable among replicates? (e.g. It is biological difference rather than technical bias.)

Minor points:

1. Have authors examined the efficiency of TMEM16F incorporation using fluorescence intensity? Is it comparable to SER5 incorporation?

Author Response

Raghunath et al reported an interesting finding that disruption of PS distribution associates with incorporated SER5 protein HIV-1 envelope but does not determine restrictive function of SER5. This may help understand the complex mechanism of HIV host restriction and potential therapeutic applications. Several points have to be addressed to have it published.

Major points:

“1. In Figure 2B-C and Figure 5 D-F, what’s the biological relevance of Pearson’s coefficient?”

We are using Pearson’s coefficient to test if there is correlation between SERINC incorporation and AnxV levels on a single particle level. In other words, by extracting intensity values from both iHA and AnxV channels, we can test whether the iHA and AnxV intensities are correlated with one another on a single-particle level. If viral incorporation of SERINC was indeed correlated with the AnxV staining (indicated by a Pearson’s coefficient value closer to 1), it implies that PS exposure in virions is dependent on the amount of SERINC proteins incorporated into the individual virions. As expected, SER5 incorporation and AnxV signal exhibit higher Pearson’s coefficients than SER2 incorporation and AnxV signal, which led us to conclude that the SER5 incorporation and PS staining are correlated. We have provided additional context regarding the use of Pearson’s correlation analysis in the revised manuscript.

“Note that in SER2 treatment most virions have lower iHA intensity, which suggests less efficient incorporation of SER2 than SER5. This raise a question whether a linear model is appropriate, since it’s difficult to assess the relationship between protein incorporation and PS externalization if the protein is less or even not incorporated.”

We thank the reviewer for pointing this out. While Figures 2B and 5D does appear to indicate poorer incorporation of SER2, this isn’t true, as evidenced by independent intensity analysis shown in Figure 2B and Figure 5D (pointed out in #2 by the reviewer). We apologize for the confusion and acknowledge that the data likely appears contradictory to most readers without this context.

This is a result of our data analysis algorithm used specifically for extracting multi-channel intensity data, which is slightly different from the single-channel data extraction algorithm used in Figures 2B and 5D. As stated in the text of previous version of the manuscript “By stringently selecting for particles that are positive for both AnxV and iHA signals, we quantitatively compared intensities of iHA vs AnxV for each virion (Figure 2C)”; we were hyper-selecting for particles that were positive for AnxV in both SER5 and SER2 viruses. Given that SER2 viruses typically exhibit lesser AnxV signal overall (Figure 1E in revised manuscript), this protocol discards a large fraction of SER2-iHA positive particles, many of which have high levels of this protein. This analysis does not considerably affect SER5 particles that generally contain higher levels of PS.

To address this caveat, we have re-analyzed images, using only fluorescence from the viral core marker (GFP-Vpr) and plotting unfiltered values of AnxV and iHA intensities from all the puncta positive for GFP-Vpr (except puncta that exhibit signal saturation and background-level signals). This updated method is agnostic to iHA and AnxV intensities.

However, owing to the unfiltered nature of the updated plots, the readability of the correlation scatterplot is affected. Despite readability concerns, we have opted to update our manuscript with a version of these figures, in hopes to not mislead the reader regarding the relative iHA and AnxV intensities between the samples. We once again thank the reviewer for bringing this to our attention.

We would additionally like to point out, that linear regression fits were used merely to guide the reader towards the trend, and not to make any quantitative claims regarding correlation. We have clarified this in the revised version of the manuscript. We are using Pearson’s coefficient analysis for comparing correlation between iHA and AnxV across multiple preparations, as shown in the revised version of the manuscript (Figure 2D and Figure 5E).

For the reviewer’s convenience, we are attaching a version of the updated plot here.

“2. Note that in Figure 1C and Figure 5C, SER2 incorporation is significantly higher than SER5 as measured by iHA fluorescence intensity. These data are not in line with Figure 2B and Figure 5D. Please consider possible reasons.”

We have addressed this criticism in the explanation for Point #1.

“3. Please include control fluorescent images for Figure 1 (Ctrl) and Figure 2 (SER2-iHA).”

We thank the reviewer for this suggestion. We have included a panel to show images for Ctrl and SER2-iHA.

"The fluorescence images are quantified as intensity which seems to have an arbitrary unit. Could author please describe how to make readings from different channels (annexin V and iHA) comparable among replicates? (e.g. It is biological difference rather than technical bias.)"

The use of arbitrary units for imaging experiments is a standard practice in the field. We maintain uniform laser powers with calibrated laser sources for all our imaging experiments. We take extra care to re-use pixel sizes, bit depth with identical spectral windows for every unique fluorophore used in the study. For image analysis, we are utilizing an open-source software (ICY) using a built-in particle detection algorithm for per-object intensity quantification. For every virus panel, care was taken to ensure that the threshold value inputs were consistent for the entire panel. For example, after imaging the samples containing Ctrl, SER5, SER2 and TMEM16F containing viruses for a panel; we made sure that identical threshold values were used for all the images by saving the analysis protocol and reusing the same settings for analyzing all the images from that panel. This ensures a level of consistency that is required to compare intensity data between different preparations. We have adequately described our image analysis workflow in Materials and Methods. For further clarification on how multi-color channel images were treated in this workflow, we have added additional context in the Materials and Methods section.

We were careful to compare the intensities of different samples within a particular panel and report the biological reproducibility for these differences. For example, every single biological replicate imaged on the same day, using the same settings, and analyzed using identical algorithm, consistently exhibits an increase in PS staining of SER5 (Figure 1E, 1F of revised manuscript) and TMEM16F (Figure 4A, 5B) viruses, relative to Ctrl and SER2 viruses. Similarly, we see a consistent increase in PS exposure for both Ctrl and SER2 viruses in samples produced from ΔCDC50a cells relative to the same panel of viruses produced in WT 293T cells (Figure 5B).

Minor points:

  1. Have authors examined the efficiency of TMEM16F incorporation using fluorescence intensity? Is it comparable to SER5 incorporation?

We appreciate this comment from the reviewer. We chose not to pursue imaging experiments involving TMEM16F incorporation, owing to the lack of a simple, solvent-accessible tag suitable for immunofluorescence (like iHA on SER5/2). While our TMEM16F construct does carry a flag-tag; the viruses might need permeabilization for immunolabeling, which further complicates the analysis.

We would also like to reiterate that the rationale behind the use of TMEM16F for this study, was to use a non-specific scramblase to achieve high PS exposure in viruses to study the specific effects of PS exposure in viral infectivity without the presence of SER5. Given that TMEM16F only leads to minimal infectivity reduction (Figure 4B) despite high PS exposure, (Figure 4A) we believe that this data further confirms the observation from a previous study (PMID: 37474505) and adds more evidence to our working hypothesis.

Reviewer 4 Report

Comments and Suggestions for Authors

The manuscript by Rhagunath et al. describes experiments to assess the relationship between SER5 and phosphatidylserine on HIV-1 restriction. The manuscript details numerous experiments including innovative approaches to address specific questions. Each question and experimental approach is clearly stated and assessed. The experiments are scientifically sound to the satisfaction of this reviewer. The manuscript is very well written and well referenced.

While the manuscript may be of average interest to the general reader, it should be of very high interest to those working in the field of virology and in the area of HIV specifically.

Author Response

“The manuscript by Rhagunath et al. describes experiments to assess the relationship between SER5 and phosphatidylserine on HIV-1 restriction. The manuscript details numerous experiments including innovative approaches to address specific questions. Each question and experimental approach is clearly stated and assessed. The experiments are scientifically sound to the satisfaction of this reviewer. The manuscript is very well written and well referenced.

While the manuscript may be of average interest to the general reader, it should be of very high interest to those working in the field of virology and in the area of HIV specifically.”

We thank the reviewer for their kind comments about our work. We hope the article will be of broad interest to virologists in general. 

Reviewer 5 Report

Comments and Suggestions for Authors

This interesting manuscript delves into the possible molecular mechanisms by which SER5 membrane protein may inhibit HIV-1 infection. In particular, the results of this study indicate that externalization of phosphatidylserine (PS) in the envelope of HIV-1 pseudoviral particles is not the main mechanism for SER5-induced host cell restriction. The study is based on a firm background, the experiments have been smartly designed and carefully performed, the results are clear and properly discussed in comparison with the current literature. Despite the original working hypothesis has not been confirmed by the results, this study contains a remarkable amount of information, also useful from a methodological point of view for the set-up of experiments with cytotropic liposomes and membrane-wrapped nanoparticles. 

I have no suggestions to improve an already excellent work.

Author Response

“This interesting manuscript delves into the possible molecular mechanisms by which SER5 membrane protein may inhibit HIV-1 infection. In particular, the results of this study indicate that externalization of phosphatidylserine (PS) in the envelope of HIV-1 pseudoviral particles is not the main mechanism for SER5-induced host cell restriction. The study is based on a firm background, the experiments have been smartly designed and carefully performed, the results are clear and properly discussed in comparison with the current literature. Despite the original working hypothesis has not been confirmed by the results, this study contains a remarkable amount of information, also useful from a methodological point of view for the set-up of experiments with cytotropic liposomes and membrane-wrapped nanoparticles. 

I have no suggestions to improve an already excellent work.”

We thank the reviewer for the kind and thoughtful comments regarding our manuscript.

Round 2

Reviewer 3 Report

Comments and Suggestions for Authors

The authors have fully addressed my points and I'd like to recommend acceptance of current manuscript.